# Design of Experiments-Based Optimization of Flavonoids Extraction from *Daphne genkwa* Flower Buds and Flavonoids Contents at Different Blooming Stages

**DOI:** 10.3390/plants11070925

**Published:** 2022-03-29

**Authors:** Min-Kyoung Kim, Geonha Park, Yura Ji, Yun-Gyo Lee, Minsik Choi, Seung-Hyeon Go, Miwon Son, Young-Pyo Jang

**Affiliations:** 1Division of Pharmacognosy, College of Pharmacy, Kyung Hee University, Seoul 02447, Korea; mindung3@khu.ac.kr; 2Department of Life and Nanopharmaceutical Sciences, Graduate School, Kyung Hee University, Seoul 02447, Korea; ginapark0326@khu.ac.kr (G.P.); j5620242@naver.com (Y.J.); 3Department of Biomedical and Pharmaceutical Sciences, Graduate School, Kyung Hee University, Seoul 02447, Korea; dbsry3733@khu.ac.kr (Y.-G.L.); alstlr7595@khu.ac.kr (M.C.); 2021310688@khu.ac.kr (S.-H.G.); 4Central Research Center, Mtherapharma Co., Seoul 07793, Korea; mwson2020@mtherapharma.com; 5Department of Oriental Pharmaceutical Science, College of Pharmacy, Kyung Hee University, Seoul 02447, Korea

**Keywords:** *Daphne genkwa*, Thymelaeaceae, flavonoids, design of experiments, blooming stages

## Abstract

The flower buds of *Daphne genkwa* have been reported as a potent resource associated with anti-angiogenic, anti-tumor, anti-rheumatoid arthritis activities, as well as immunoregulation. This paper aimed to establish an optimal extraction method for flavonoids, as active phytochemicals, and to conduct a comparative analysis by profiling the different blooming stages. Optimized shaking extraction conditions from the design of experiments (DoE), such as minutely mixture design, 2^3^ full factorial design, and polynomial regression analysis, involved an agitation speed of 150 rpm and temperature of 65 °C for 12 h in 56% (*v*/*v*) acetone solvent. After, a comparative analysis was performed on three blooming stages, juvenile bud, mature purple bud, and complete flowering, by ultra-high-performance liquid chromatography-photodiode array-mass spectrometry (UHPLC-PDA-MS). Most flavonoids increased during bud growth and then decreased when the bud opened for blooming. In particular, apigenin 7-*O*-glucuronide, genkwanin 5-*O*-primeveroside, and genkwanin strikingly showcased this pattern. Furthermore, the raw spectrometric dataset was subjected to orthogonal projection to latent structures discriminant analysis (OPLS-DA) to find significant differences in the flavonoids from the juvenile bud, mature purple bud, and complete flowering. In conclusion, the present study facilitates an understanding of flavonoid change at different blooming stages and provides a momentous reference in the research of *D. genkwa*.

## 1. Introduction

Botanical medicine development, including interest in their health benefit and/or ability to protect our body from health disorders, is increasing worldwide [1]. The flower buds of *Daphne genkwa* Siebold & Zucc. (Thymelaeaceae) is a well-known traditional oriental medicine and are widely used in Korea and China. *D. genkwa* was found to contain important secondary metabolites including flavonoids, lignans, terpenoids, and coumarins that are actively studied for various biological activities [2,3]. In Eastern Asia, flower buds of *D. genkwa* (Genkwae Flos) are traditionally used as medicinal parts and have been studied for various pharmacological activities. Many studies have suggested daphnane-type diterpenes as active ingredients that are responsible for diverse biological activities such as antineoplastic, neuroprotective, anti-human immunodeficiency virus (HIV), etc. [4,5,6,7,8]. Phytochemical analysis showed that the flower buds contain relatively smaller amounts of diterpenes than other plant parts such as roots and stems, and other parts were also studied for the biological activities [9]. Meanwhile, recent studies have shown that Genkwae Flos, which contain flavonoids as major components [10], exhibit anti-angiogenetic, anti-inflammatory [11], anti-rheumatoid arthritis [12], and antitumor [13,14] activities. The analysis of flavonoids, which are major secondary metabolites of Genkwae Flos, is important to establish an efficient quality control strategy for Genkwae Flos as a medicinal raw material.

To the best of our knowledge, there is no research focusing on the comparative analysis for the classification of blooming stages of *D. genkwa* pertaining to the flavonoids profile. Indeed, flavonoids are very rich and primarily affected by the growth stage of the flower in diverse species such as *Chrysanthemum morifolium*, *Lycoris radiata*, *Agastache rugosa*, etc. [15,16,17]. Likewise, if Genkwae Flos shows a large difference in flavonoids content during the flowering period, this could be an important quality variable in the development of medicines using Genkwae Flos as a raw material. This study was conducted to provide scientific evidence for these questions.

According to the characteristic morphology and color, the medicinal part of *D. genkwa* flowers could be divided into three stages: juvenile bud (S1), mature purple bud (S2), and complete flowering (S3). As can be seen in Figure 1, the juvenile buds (S1) are green and densely covered with white downy hair. The mature purple buds (S2) are fully purple and the hair on the top has disappeared. Finally, at the complete flowering (S3), blooming purple flowers are gathered in an umbel [18].

Our previous study [19] established a systematic and robust ultra-high-performance liquid chromatography-photodiode array (UHPLC-PDA) analysis that could identify and quantify eleven flavonoids of *D. genkwa*: (1) apigenin 5-*O*-glucoside, (2) apigenin 7-*O*-glucoside, (3) yuanhuanin, (4) apigenin 7-*O*-glucuronide, (5) genkwanin 5-*O*-primeveroside, (6) genkwanin 5-*O*-glucoside, (7) genkwanin 4′-*O*-rutinoside, (8) tiliroside, (9) apigenin, (10) 3′-hydroxygenkwanin, and (11) genkwanin. The representative chromatographic profile is shown in Figure 2 and eleven flavonoid peaks are indicated, with accompanying structural information. In this study, we tried to reveal the relationship between changes in the flavonoid profile and the three blooming stages using metabolomics technology.

Prior to the metabolomics analysis, the optimized extraction solvents and conditions showing the maximum flavonoid extraction efficiency were needed to be optimized by design of experiments (DoEs) for reliable and consistent metabolomics studies. DoE is used to plan to-be-performed experiments for approaching the question to be solved and considers how to analyze the data using statistical methods in order to gather the maximum amount of information from a minimum number of experiments [20]. Experiments are conducted according to the experimental design and the responses are recorded for determining main effects and interaction effects between multi-variables. To extract flavonoids from plant matrix, solvent properties are very important to penetrate plant cell walls and dissolve intracellular metabolites. In this regard, a mixed composition of water and organic solvents was used for the extraction of phytochemicals with diverse degrees of polarity [21,22,23]. Therefore, the optimization of ternary mixtures was investigated with the mixture design method to test extraction efficacy in totality from non-polar aglycones (genkwanin and apigenin) to polar flavonoid glycosides.

After defining the optimal solvent mixture system, the best shaking extraction conditions were explored using 2^3^ full factorial design (FFD) and polynomial regression analysis. Shaking extraction could minimize the degradation of the heat-sensitive natural products and consumes low energy, which is particularly advantageous for large-scale industrial processes [23,24]. The parameters that could affect the extract efficiency of the shaking procedure are agitation speed, extraction time, setting temperature, and solvent-to-material ratio [25]. Since these parameters influence each other and a complex interaction can be observed, DoE is widely used to evaluate multivariable effects based on statistical analysis [26]. Among the various approaches to design the experiments, FFD is utilized because it enables to discuss the joint effect of the studied factors (or design parameters) on a response with all possible combinations of levels for all factors [26]. In this study, the significant parameters for shaking extraction were elucidated through FFD and the best condition was determined by polynomial regression analysis.

## 2. Results

### 2.1. Optimization of the Extraction Solvents

To penetrate the complex plant matrix, aqueous mixtures with alcoholic solvents, acetone, and ethyl acetate were considered suitable for diverse plant sources [27]. Also, flavonoids have polar (glycosidic form) to less polar (aglycone form, mainly flavones) constituents and predominant secondary metabolites in *D. genkwa* [19]. Therefore, water and two polar organic solvents, a dipole moment (μ) of >2 (acetone 2.69 and methanol 2.87), were selected considering water-mixability.

Designing an approach to understand the best solvent for enriching target metabolites, flavonoids, we used the mixture design with the primary three solvents being selected: water (X_1_), acetone (X_2_), and methanol (X_3_). When performing the mixture design, the shaking extraction process was at 200 rpm, 22 ± 2 °C, for 24 h, and in 30 mg/mL.

A total of 12 assays (Table 1) using the simplex-centroid design were executed and each response function was expressed as total flavonoids extraction efficacy, calculated by summing all identified flavonoid peak areas (Figure 2). The independent 12 assays were all performed in triplicate to evaluate the experimental error. The *p*-value (0.001) showed strong evidence that the model was very suitable. Also, the high R^2^ value (95.94%) and lack of fit (0.158) validated that all response functions fit the experimental data well with good predictability in the extraction of total flavonoids using various solvent mixtures (Table 2).

According to the analysis of regression coefficients at the 95% confidence level, the linear terms X_1_, X_2_, and X_3_ and the interaction terms X_1_X_2_ and X_1_X_3_ were significant whereas the interaction term X_2_X_3_ was not significant but contributed to the fitness of the model, therefore the following equation was mathematically calculated:Y = −66788·*X_1_ + 2078100·*X_2_ + 5355400·*X_3_ + 19773000·*X_1_X_2_ + 1275800·*X_1_X_3_ + 3122300·X_2_X_3_(1)
where Y is expressed in summing eleven identified flavonoids peak areas, * means significant parameters (*p* < 0.05), X_1_ is water, X_2_ is acetone, and X_3_ is methanol.

The best solvent mixture was calculated by response surface methodology using upper multivariate Equation (1) by illustration with a 3D ternary graph (Figure 3A) and 2D contour plot (Figure 3B). In the 3D ternary graph (Figure 3A), the optimal zone where flavonoids are maximally extracted is shown at the top of the non-symmetrical open plane curve formed by interaction with acetone (X_2_) and water (X_1_). As can be seen from the optimization diagram (Figure 3C), both acetone (X_2_) and water (X_1_) have parabolic curve influence on the flavonoids extraction efficacy that can calculate the response of the maximum value. Methanol (X_3_) also has a parabolic curve effect but an opposite shape, and does not even contain an optimal mixture solvent point, which is indicated by a red line in each solvent of optimization diagram (Figure 3C).

Finally, a binary mixture consisting of water (X_1_) and acetone (X_2_) in a 44:56 ratio (X_1_:X_2_) achieved the best extraction and yielded a Y value of 600,700 as shown in the optimization diagram (Figure 3C).

### 2.2. Optimization of the Shaking Extraction Process

After the preliminary study for the solvent system, where the optimized solvent mixture was defined as 56% acetone in water, the detailed shaking procedure was optimized. The purpose of extraction is to efficiently separate the soluble metabolites of plants using selective solvents through appropriate procedures [27]. Among the various extraction methods, the shaking method is an economic and low-energy consumption extraction process that is often adopted in the natural products industry, and is very practical to yield high amounts of phytochemicals [28]. To optimize the shaking extraction conditions for maximum extraction of flavonoids, two stages of optimization were pursued: (a) a screening step, where three variables (extraction time, extraction temperature, shaking speed) were studied to identify those with significant effects on the dependent variable (flavonoid extraction efficacy), and (b) the final optimization step, where the significant variable was further examined to determine the best condition. The two-level FFD is well-known for screening design to select meaningful factors and understand individual and/or synergetic effects in a multivariable system [29].

A 2^3^ FFD was used to screen three variables such as extraction time (X_a_), extraction temperature (X_b_), and shaking speed (X_c_), which have an important influence on the extraction efficiency of the shaking method [28]. A total of eleven runs with three replicates in the central point (Table 3) was given by the expression *2k* + *C*, where *k* is the number of variables, and *C* is the number of central points [30], and all points were performed in triplicate for the evaluation of error. An ANOVA (Table 2) showed that shaking temperature (X_b_) and the interaction of shaking time and shaking speed (X_a_∙X_c_) were significant with a great coefficient to the efficacy of flavonoids extraction evidenced by small *p*-values (<0.05). In general, shaking speed and extraction time have a significant effect on the extraction efficiency of target metabolites, but in the case of plant samples with fragile tissues such as Genkwae Flos, these parameters do not seem to have a significant effect. Among the parameters, the extraction temperature (X_b_) showed a significant influence compared to the X_a_∙X_c_ interaction which is shown from the much higher F-values. Also, the high *R*^2^ value (94.75%) and low *p*-value (0.001) showed good evidence of suitability and accuracy of this experimental model to predict the response. As can be seen in the Pareto chart (Figure 4A), main effect diagram (Figure 4B), and interaction effect diagram (Figure 4C), the X_b_ and X_a_∙X_c_ had a striking positive influence on the flavonoids extraction efficacy and the temperature (X_b_) bar has the longest length in the Pareto chart (Figure 4A), which means this effect is the greatest. Those results were also summarized by the following equation:
Y = 7872868 + 5206·X_a_ + 24666·X_b_ + 1628·X_c_ − 91.2·X_a_X_b_ − 39.4·*X_a_X_c_ − 35.4·X_b_X_c_ + 0.595·X_a_X_b_X_c_(2)
where Y is expressed as the sum of the eleven identified flavonoids peak area, * means significant parameters (*p* < 0.05), X_a_ is shaking time, X_b_ is shaking temperature, and X_c_ is shaking speed.

For the optimization step, the most influencing factor of temperature (X_b_) was further analyzed to define the best extraction condition. The designed X_b_ had four steps ranging from 20 to 80 °C considering the boiling point (70.2 °C) of the solvent mixture calculated from the mole % acetone in water [31]. Each of four experimental points was performed in triplicate and ANOVA analysis was performed to describe statistical suitability. To avoid overfitting, stepwise regression analysis (first-order, second-order, and third-order equations) was performed and compared the goodness of fit of the data based on two calculated evaluation measures, Akaike Corrected Information Criterion (AICc) and Bayesian Information Criterion (BIC) (Table 4). The smaller the AICc and BIC are, the better the model fits the data. As shown in Table 4, in the third-order equation model, AICc was the lowest at 851.65, and BIC was also the lowest at 857.57, which indicates the best fit for the data. The high values of the regression coefficient (*R*^2^ = 95.85%) and a high level of statistical significance (*p*-value < 0.001) also validated the selection of the third-order model and high degree of data fitness (Table 4). The third-order regression equation is given as follows:Y = 8672000 − 23670·X_b_ + 860·X_b_^2^ − 6.905·X_b_^3^(3)
where Y is defined as the sum of peak areas of the identified eleven flavonoids and X_b_ is temperature.

By mathematical approach, the maximum point of the three-order polynomial equation was found at 65.6 °C (Figure 5). Finally, the optimum shaking extraction conditions to maximize the flavonoids extraction from *D. genkwa* flower buds were determined as 44:56 ratio (water: acetone, *v*/*v*) solvent mixture, 150 rpm shaking speed, 12 h shaking time, and 65 °C temperature. To validate this model, six replicates were extracted with the optimal conditions. The experimental value was 8,955,360 ± 36,450 (*N* = 6), which was close to the predicted value (8,871,048) with 0.94% percent error, sufficiently satisfying the predictability of the model.

### 2.3. Flavonoids Profile Change by Blooming Stages

In order to analyze the change in the content of individual flavonoid components according to the growth period of the buds (juvenile bud (S1), mature purple bud (S2), and complete flowering (S3)), the area values for the identified 11 flavonoid peaks were analyzed from the samples of the three different stages. A total of 108 chromatographic raw data (from triplicate of 36 samples) were converted into binary format and introduced to SIMCA-P for the multivariate analysis. Then Orthogonal Partial Least Squares Discriminant Analysis (OPLS-DA) was performed to investigate whether the discrimination of growth stages was possible in terms of the changes in flavonoid profiles. OPLS is an extension of PLS which can separate the systematic variation of data matrix into classes by finding important variables that maximize the variance between classes and also orthogonal latent variables to separate by classes [32]. In the present study, a score plot (Figure 6A) showed a clear clustering of three flowering stages with classification rate of 100%. The model was quite consistent and robust by the values of *R*^2^X (0.982) and *R*^2^Y (0.928) parameters. Also, the high value of *Q*^2^Y (0.901) indicates great predictability. Besides, CV-ANOVA (*p* < 0.001), Hotelling’s T^2^ using 95% and 99% confidence limits, and permutation testing (*N* = 200) were performed to exclude overfitting and validate the model (Figure 6B,C).

In order to elucidate flavonoid component(s) responsible for the differentiation of each flowering stage, the Variable Importance in Projection (VIP) approach was applied. Three flavonoids possessing the highest VIP score > 1.000 (adjusted *p* < 0.5) were selected as follows; apigenin 7-*O*-glucuronide (3.574), genkwanin 5-*O*-primeveroside (1.178), and genkwanin (1.208). 

Afterwards, a heatmap of eleven flavonoids contents according to three stages was visualized (Figure 7A). The heatmap for three flavonoids of apigenin 7-*O*-glucuronide, genkwanin 5-*O*-primeveroside, and genkwanin were extracted separately because they showed a similar pattern of contents change according to the blooming stages. The relative content chart for these three flavonoids was represented to show the changing trend during growing stages (Figure 7B). These components increased when the bud matured (S1 to S2) and decreased when fully bloomed (S2 to S3).

## 3. Discussion

The quality control of botanical raw material is gradually progressing in the direction of identification and quantification of all possible phytochemicals using chemical fingerprinting and profiling approaches [33]. Optimization of the extraction process in the analysis of plant raw materials is the most basic step that must precede all analysis studies. Therefore, the best solvent mixture system and shaking extraction conditions for the maximum extraction of flavonoids from *D. Genkwa* flower buds were optimized in this study by various DoEs. The metabolomics studies using the optimized extraction conditions could greatly improve the robustness of the experimental results. It is interesting that the shaking speed and extraction time, which are factors affecting the extraction efficiency of the shaking extraction method in general, have a lower effect than the extraction temperature in the case of Genkwae Flos. This is presumed to be due to the characteristics of the tissue of Genkwae Flos, which is softer and weaker than other plant parts, so that it is easily penetrated by solvents. Next, a series of metabolomics studies were performed to clarify whether flavonoid components, the main ingredient of Genkwae Flos, change by flowering stages, which will provide important data for the quality control of drug products developed with Genkwae Flos as a raw material. As a result, most flavonoids increased as the buds grew and then decreased when the buds were in full bloom (Appendix A). In particular, this pattern is clearly observed in three unique flavonoids, apigenin 7-*O*-glucuronide, genkwanin 5-*O*-primeveroside, and genkwanin, which are also selected as the most influencing components for the discrimination of flowering stages by multivariate analysis. If further research is conducted on whether these components are deeply correlated with the biological activities of Genkwae Flos [10,11,12,13,14], it will be possible to know when it is most efficient to harvest the material using the information established through this study. As in this study, plant metabolomics studies systematically conducted through the design of experiments will be of great help in developing more accurate and reproducible quality control technologies for medicinal herbal products.

## 4. Materials and Methods

### 4.1. Chemicals and Reagents

Extra pure grade solvents (purity > 99.5%) of methanol, acetone, and water were supplied from Duksan Pure Chemicals (Seoul, Korea). For UHPLC-PDA analysis, HPLC-grade solvents (purity > 99.9%) of acetonitrile and water and HPLC-grade formic acid (purity > 99.0%) were purchased from Thermo Fisher Scientific (Seoul, Korea). 

### 4.2. Plant Material

The buds or flowers of *D. genkwa* at three different blooming stages, juvenile bud (S1), mature purple bud (S2), and complete flowering (S3), were collected from 12 Genkwa trees in Korea in April to June 2019. A dozen of biological replicates (A to L) were selected in this study, of which five (A to E) were collected from Medicinal Herb Garden of College of Pharmacy, Kyung Hee University, Seoul, and the other seven (F to L) from greenhouses in different places. The botanical origin of all trees (A to L) was authenticated by morphological examination including leaf arrangement, flower color, shape of leaf, number of stamens, inflorescent, etc., compared to the previous taxonomic literature [18,19]. The collected buds or flowers were immediately lyophilized and labeled as DG-A-S1-2019 to DG-L-S3-2019 (Appendix A). The samples were stored in a deep freezer until analyses.

### 4.3. Sample Preparation

Thirty milligrams of each lyophilized sample and 5 mm steel bead (QIAGEN) were placed in a 2 mL tube (Eppendorf) and ground to homogeneous fine powder by Tissue-Lyser (QIAGEN) in the frequency of 30 Hz for 3 min. For the comparative analysis in different blooming stages, those milled samples were added to 1 mL of the solvent established through mixture design and extracted according to the optimal shaking extraction conditions from the 2^3^ full factorial design and polynomial regression analysis.

### 4.4. UHPLC-PDA Analysis

Flavonoid identification was performed on a Waters AQCUITY^TM^ H-class UPLC system with PDA detector and operated by Empower-3 software (Waters Corporation, Milford, MA, USA). A column Kinetex-C18 (2.1 × 50 mm, 1.7 μm, Phenomenex) was used for separation, with the column temperature set at 28 °C. For the mobile phase, the binary gradient elution system consisted of acetonitrile (A) and 0.1% (*v*/*v*) formic acid in water (B). The elution method was as follows: linear gradient from 10 to 45% A (0–13 min), 45 to 100% A (13–13.5 min), 100 to 10% A (13.5–14 min), and isocratic 10% A (14–15 min). The flow rate and sample manager temperature were 0.35 mL/min and 25 °C, respectively. The detection wavelength was fixed at 335 nm. Those UHPLC conditions were optimized by analytical quality-by-design (AQbD) and validated in our previous report [25].

### 4.5. Mixture Design

A mixture design was used to find the optimum solvent composition for the maximum flavonoids extraction. Twelve representative mixtures were set up by simplex centroid design (Table 1) using water, acetone, and methanol as follows: the triangle vertices at 1:0:0 ratio (*v*/*v*/*v*) as pure solvents; the edges at 1/2:1/2:0 ratio (*v*/*v*/*v*) as binary mixtures; three central points to estimate the pure error at 1/3:1/3:1/3 ratio (*v*/*v*/*v*) as equivalent ternary mixture; the axial points at 2/3:1/6:1/6 ratio (*v*/*v*/*v*) as non-equivalent ternary mixture. For the study of optimized solvents system, the extraction conditions were set as shaking speed of 200 rpm for 24 h at room temperature (22 ± 2 °C). All twelve points were performed in triplicate for the evaluation of experimental error. The content of total flavonoids was the dependent variable, which was obtained by summing eleven flavonoid peaks areas in each chromatographic result. The Minitab software ver. 18 and Statistica software ver. 13.3.0 (TIBCO Software Inc., Palo Alto, CA, USA) were employed for experimental design, statistical analysis, and model building. The following second-order polynomial Equation (4) represents this model:Y = *β*_0_ + ∑_1≤*i*≤3_*β_i_x_i_* + ∑_1≤*i*<*j*≤3_*β_ij_x_i_x_j_* + ∑_1≤*i*<*j*<*k*≤3_*β_ijk_x_i_x_j_x_k_*(4)
where Y is the response, *β*_0_ is the corresponding coefficient for intercept, *β_i_* is for each linear effect term, and *β_ij_* and *β_ijk_* are binary and ternary interaction effect terms. The independent variables are represented in the equation as *x_i_*, *x_j_*, and *x_k_*.

### 4.6. Optimization of the Shaking Extraction Condition

To identify significant variables, an FFD of two levels (2^3^) was applied as shown in Table 3 for shaking time (h), temperature (°C), and speed of agitation (rpm). The dependent variable was the same as stated in the mixture design. Total experimental runs were eleven including triplicate of central point to estimate synergetic interactions in a multivariable system and to also estimate the significant factors deducing model coefficients. The results were fitted to a first-order Equation (5):Y = *β*_0_ + ∑_*i*_*β_i_x_i_* + ∑*_i_*_≠*j*_*β_ij_x_i_x_j_*(5)
where Y is the response, *x_i_* and *x_j_* are independent variables, and *β*_0_, *β_i_*, and *β_ij_* are the corresponding coefficients for intercept, linear, and interaction terms, respectively.

After elucidation of the significant variable (temperature), further polynomial regression analysis was conducted to define the best extract conditions. The experimental points were four steps ranging from 20 °C to 80 °C. All the experimental designs were performed by Minitab software ver. 18, and an ANOVA was performed for statistical suitability.

### 4.7. Multivariate Statistical Analysis

Multivariate statistical analysis was performed using SIMCA-P 14.0 (Umetrics, Malmo, Sweden). The acquired raw metabolomics dataset was elaborated for OPLS-DA study. The Hotelling’s T^2^ in 95% and 99% confidence limits were tested to investigate outliers. Cross-validation ANOVA (*p* < 0.001) was carried out to define goodness-of-fit in *R*^2^X, *R*^2^Y, and *Q*^2^Y. Permutation testing (*N* = 200) was applied to exclude model overfitting. The VIP was conducted to select marker phytochemicals having the highest classification potential. When constructing the heatmap, raw data was rescaled from 0 to 1 by variable transformation to remove any bias in actual concentration levels.

## Figures and Tables

**Figure 1 plants-11-00925-f001:**
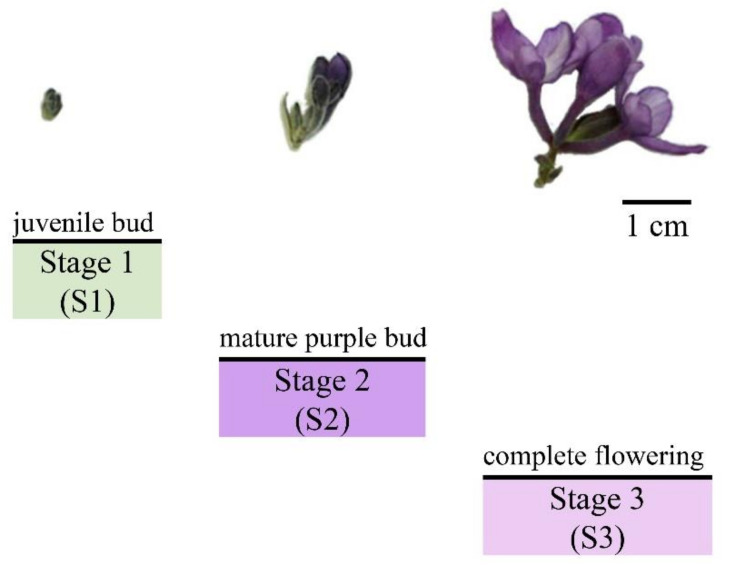
Morphological changes in the buds to flowers of *D. genkwa* at three blooming stages.

**Figure 2 plants-11-00925-f002:**
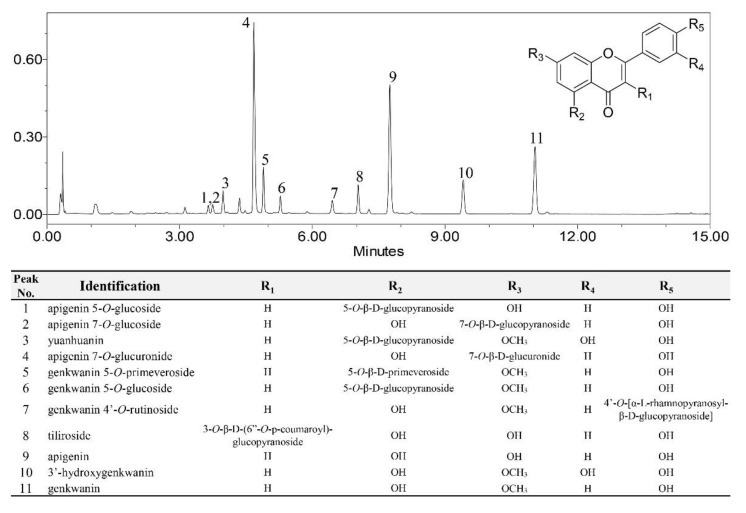
Representative UHPLC chromatogram of *D. genkwa* buds in 56% (*v*/*v*) acetone solvent. Flavonoids are indicated in the figure as 1–11: (1) apigenin 5-*O*-glucoside, (2) apigenin 7-*O*-glucoside, (3) yuanhuanin, (4) apigenin 7-*O*-glucuronide, (5) genkwanin 5-*O*-primeveroside, (6) genkwanin 5-*O*-glucoside, (7) genkwanin 4′-*O*-rutinoside, (8) tiliroside, (9) apigenin, (10) 3′-hydroxygenkwanin, and (11) genkwanin.

**Figure 3 plants-11-00925-f003:**
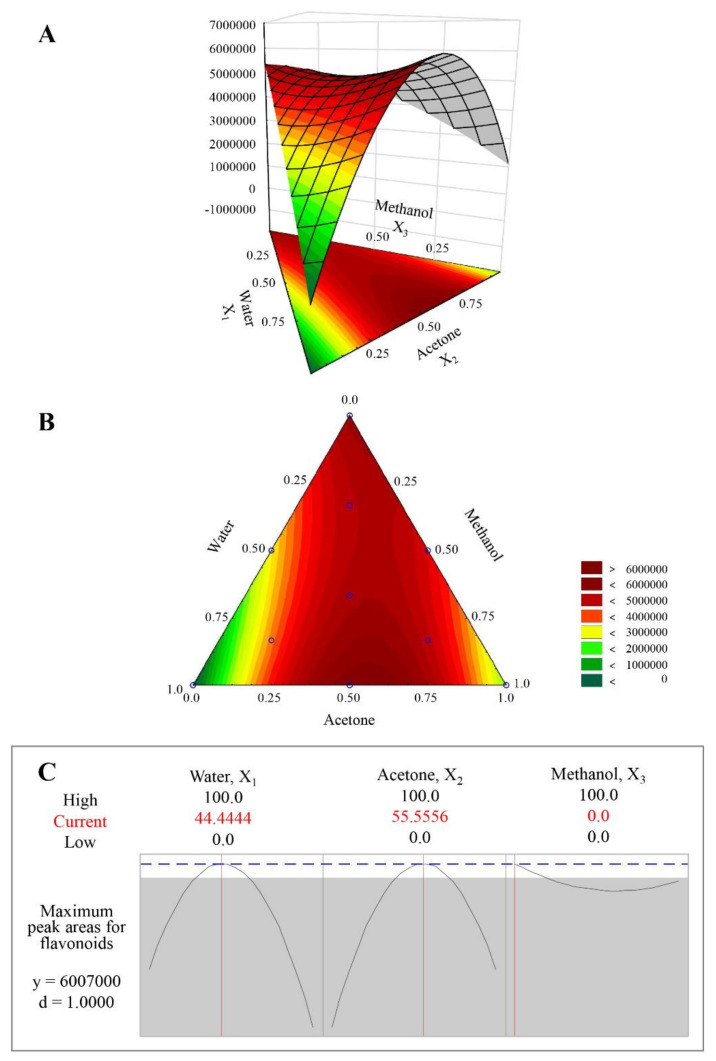
(**A**) 3D Response surface, (**B**) 2D Contour plot, and (**C**) optimization diagram obtained from mixture design for the solvent mixture of maximum flavonoids extraction efficacy.

**Figure 4 plants-11-00925-f004:**
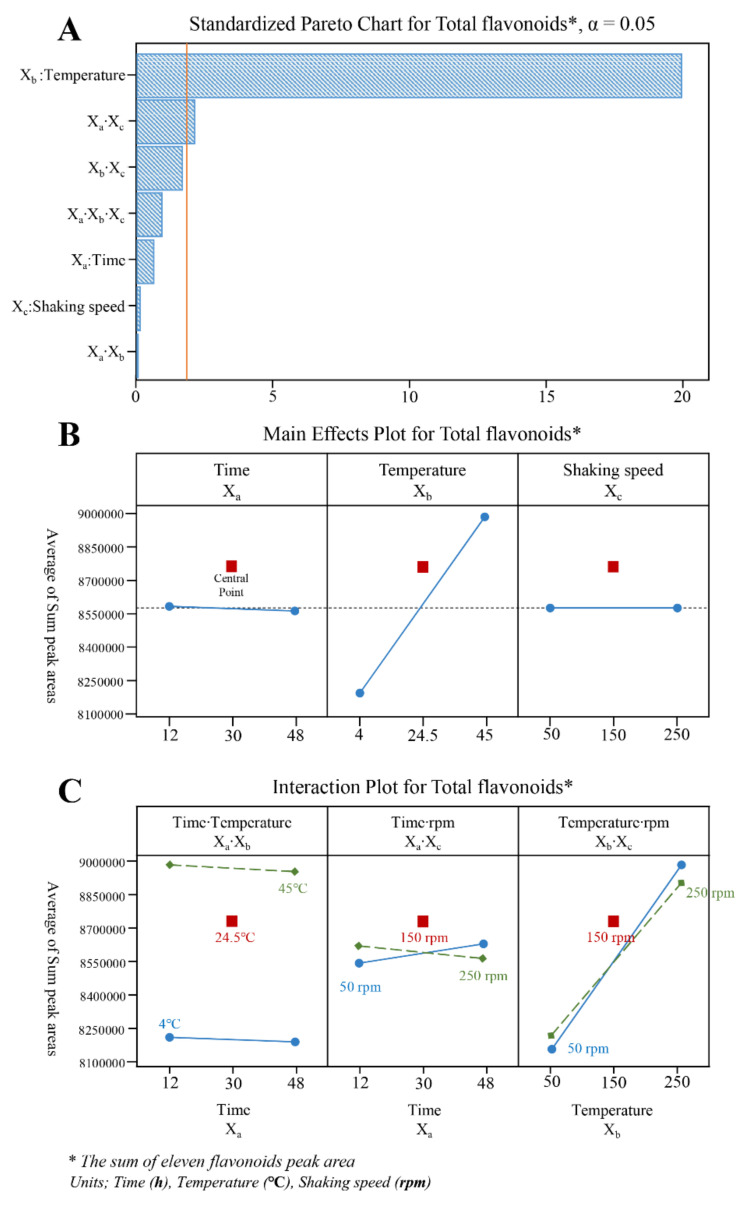
(**A**) Pareto Chart, (**B**) Main Effects Plot, and (**C**) Interaction Plot in 2^3^ FFD for optimization of flavonoids shaking extraction conditions.

**Figure 5 plants-11-00925-f005:**
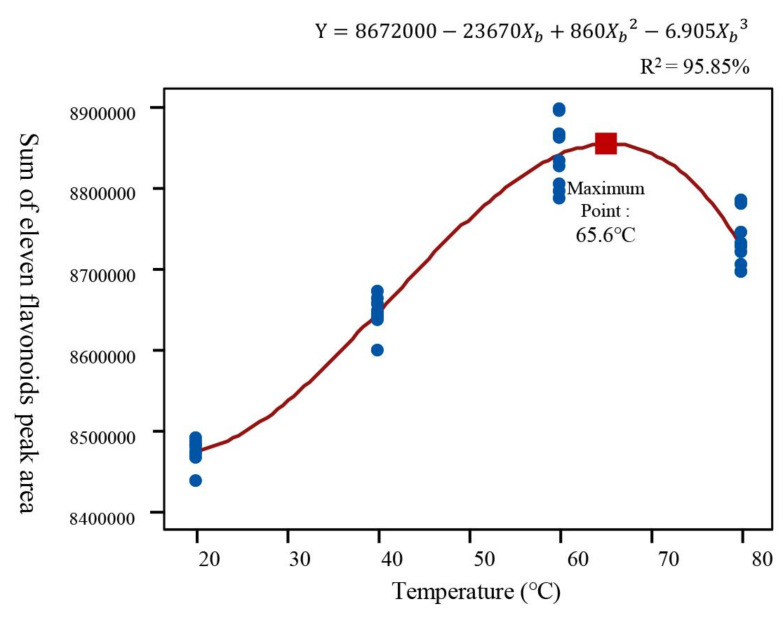
Polynomial regression curve by temperature (°C, X_b_) for the maximum flavonoids extraction efficacy.

**Figure 6 plants-11-00925-f006:**
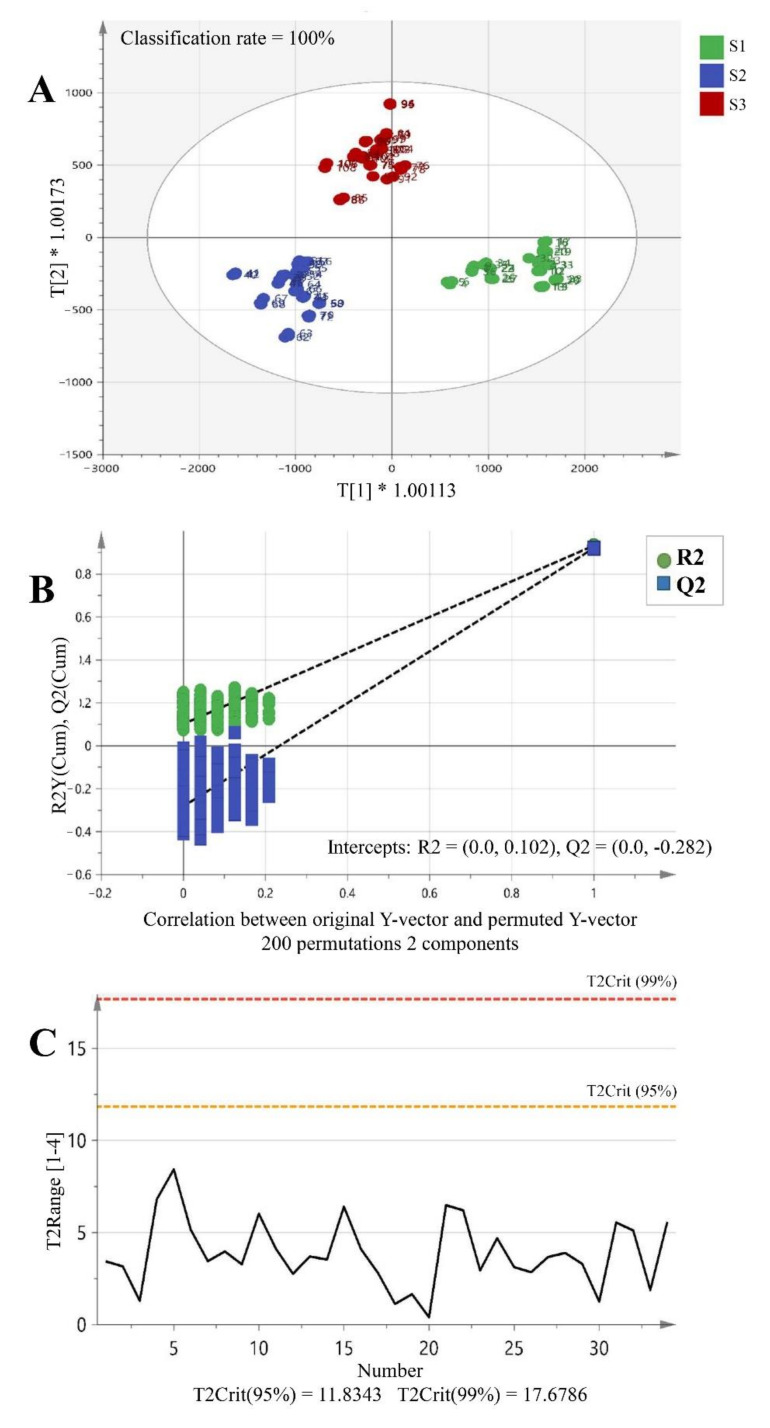
OPLS-DA score plot of samples from three stages (S1, S2, and S3) (**A**) and the validation of model using permutation test (**B**) and (**C**) Hotelling’s T^2^ test (*R*^2^X = 0.982, *R*^2^Y = 0.928, *Q*^2^Y = 0.901). T[1]*1.00113 and T[2]*1.00173 mean T[1] × 1.00113 and T[2] × 1.00173, respectively.

**Figure 7 plants-11-00925-f007:**
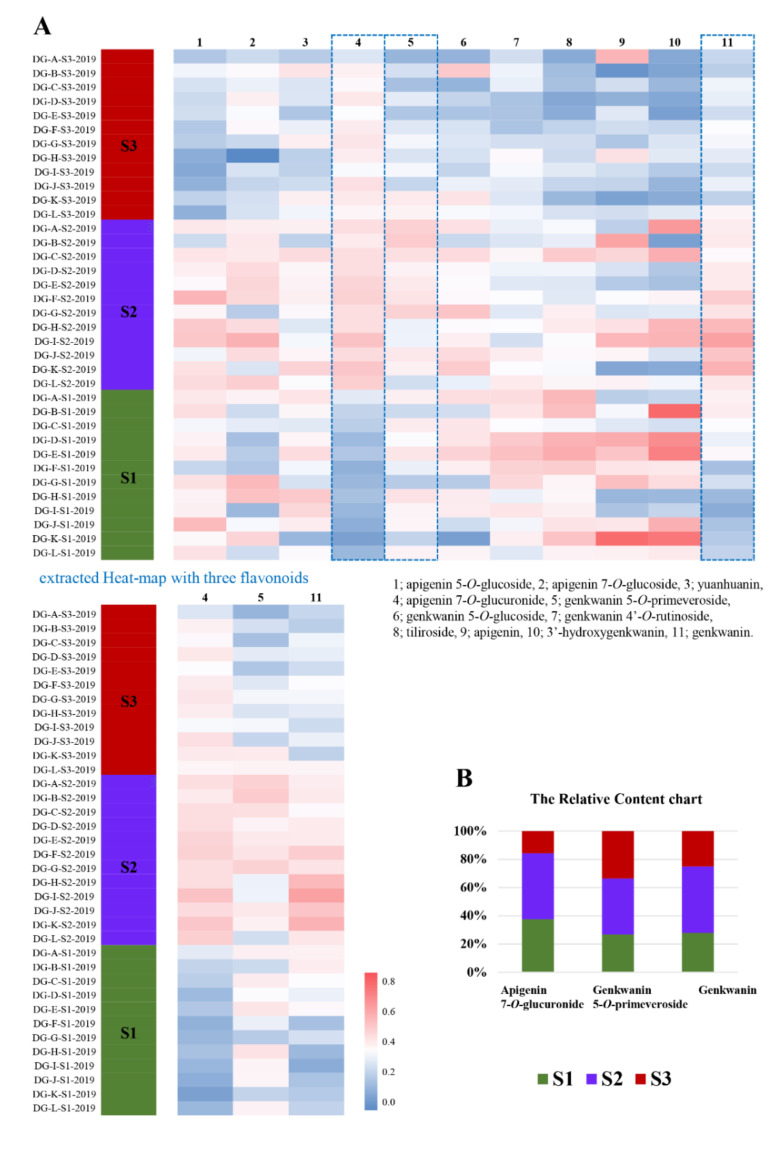
Significant changes in flavonoid biomarker candidates in S1, S2, and S3 blooming stages: (**A**) Heat-map visualization for 11 flavonoids and three significant flavonoids and (**B**) the relative content chart for three flavonoids.

**Table 1 plants-11-00925-t001:** Matrix of three variables in mixture design (simplex centroid) and experimental data for solvent mixture of maximum flavonoids extraction efficacy.

Runs	Variables			Total Flavonoids ^1^
	Water X_1_ % (*v*/*v*)	Acetone X_2_ % (*v*/*v*)	Methanol X_3_ % (*v*/*v*)	
1	100 (1)	0 (0)	0 (0)	146,058 ± 43,563
2	0 (0)	100 (1)	0 (0)	1,989,632 ± 31,687
3	0 (0)	0 (0)	100 (1)	5,451,037 ± 49,792
4	50 (1/2)	50 (1/2)	0 (0)	5,503,924 ± 33,794
5	50 (1/2)	0 (0)	50 (1/2)	2,702,388 ± 96,492
6	0 (0)	50 (1/2)	50 (1/2)	3,935,165 ± 81,567
7	66.7 (2/3)	16.7 (1/6)	16.7 (1/6)	3,330,635 ± 91,563
8	16.7 (1/6)	66.7 (2/3)	16.7 (1/6)	5,460,748 ± 78,409
9	16.7 (1/6)	16.7 (1/6)	66.7 (2/3)	5,005,935 ± 70,199
10	33.3 (1/3)	33.3 (1/3)	33.3 (1/3)	5,103,037 ± 93,384
11 (CP ^2^)	33.3 (1/3)	33.3 (1/3)	33.3 (1/3)	5,559,679 ± 69,015
12	33.3 (1/3)	33.3 (1/3)	33.3 (1/3)	5,451,037 ± 77,357

^1^ The sum of eleven flavonoids peak area, Mean ± Standard Deviation; ^2^ Central Point.

**Table 2 plants-11-00925-t002:** ANOVA for the mixture design and two-level for three variables (2^3^) FFD models.

Sources of Variations	Degree of Freedom	Sum of Squares	Mean Square	F-Value	*p*-Value
**Mixture design for solvent mixture of maximum flavonoids extraction efficacy**					
* Model	5	3.269 × 10^13^	6.538 × 10^12^	28.39	0.001
Total Error	6	1.382 × 10^12^	2.303 × 10^11^		
Lack of Fit	4	1.268 × 10^12^	3.170 × 10^11^	5.57	0.158
Pure Error	2	1.138 × 10^11^	5.690 × 10^10^		
Total Adjusted	11	3.407 × 10^13^			
**2^3^ FFD for optimization of flavonoids shaking extraction conditions**					
* Model	8	4.012 × 10^12^	5.015 × 10^11^	54.13	0.001
Time (X_a_)	1	4.458 × 10^9^	4.458 × 10^9^	0.48	0.495
* Temperature (X_b_)	1	3.757 × 10^12^	3.757 × 10^12^	405.51	0.001
Shaking speed (X_c_)	1	5.442 × 10^7^	5.442 × 10^7^	0.01	0.940
X_a_∙X_b_	1	1.173 × 10^7^	1.173 × 10^7^	0.00	0.972
* X_a_∙X_c_	1	4.806 × 10^10^	4.806 × 10^10^	5.15	0.032
X_b_∙X_c_	1	3.102 × 10^10^	3.102 × 10^10^	3.35	0.080
X_a_∙X_b_∙X_c_	1	1.158 × 10^10^	1.158 × 10^10^	1.25	0.275
Pure Error	24	2.223 × 10^11^	9.264 × 10^9^		
Total Adjusted	32	4.234 × 10^12^			

* Significant.

**Table 3 plants-11-00925-t003:** Matrix of 2^3^ FFD and experimental data for optimization of flavonoids shaking extraction conditions.

Runs	Experimental Values			Coded Values			Total Flavonoids ^1^
	Time X_a_ h	Temp. X_b_ °C	Speed X_c_ rpm	Time X_a_ h	Temp. X_b_ °C	Speed X_c_ rpm	
1	12	4	50	−1	−1	−1	8,081,710 ± 156,685
2	48	4	50	1	−1	−1	8,189,275 ± 95,743
3	12	45	50	−1	1	−1	8,990,219 ± 72,030
4	48	45	50	1	1	−1	9,007,132 ± 38,952
5	12	4	250	−1	−1	1	8,290,059 ± 155,656
6	48	4	250	1	−1	1	8,130,775 ± 65,101
7	12	45	250	−1	1	1	8,966,887 ± 100,054
8	48	45	250	1	1	1	8,892,662 ± 140,943
9	30	24	150	0	0	0	8,685,553 ± 57,434
10 (CP ^2^)	30	24	150	0	0	0	8,719,852 ± 40,179
11	30	24	150	0	0	0	8,769,795 ± 45,904

^1^ The sum of eleven flavonoids peak area, Mean ± Standard Deviation; ^2^ Central Point.

**Table 4 plants-11-00925-t004:** Stepwise regression analysis along with coefficients of each equation and model parameters.

	First-Order		Second-Order		Third-Order	
	Coefficient	*p*-Value	Coefficient	*p*-Value	Coefficient	*p*-Value
Constant	8,443,470	-	8,091,816	-	8,672,000	-
X_b_	4881	0.0001	22,463	0.0001	−23,670	0.001
X_b_^2^	-	-	−175.8	0.0001	860	0.0001
X_b_^3^	-	-	-	-	−6.905	0.0001
S	86,755.6		48,564.3		29,786.3	
R^2^	62.62%		88.63%		95.85%	
R^2^(adj)	61.52%		87.94/%		95.46%	
^1^ AICc	925.56		885.25		851.65	
^2^ BIC	929.56		890.29		857.57	

^1^ AICc; Akaike Corrected Information Criterion; ^2^ BIC; Bayesian Information Criterion.

## Data Availability

The data is available from the authors upon request.

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
