# Peer review of "Design of Experiments-Based Optimization of Flavonoids Extraction from Daphne genkwa Flower Buds and Flavonoids Contents at Different Blooming Stages"

_plants, 2022, doi:10.3390/plants11070925_

Round 1

Reviewer 1 Report

The subject of the manuscript Design of Experiments-based Optimization of Flavonoids Ex-traction from Daphne genkwa Flower Buds and Flavonoids Contents at different Blooming Stages is novelty and very interesting.

The authors show that D. Genkwa flowering stages the characterization of flavonoids was experimented in multivariate statistical analysis by extraction with those optimized conditions. The results demonstrated that most flavonoids increased as the buds growing and then decreased through the buds blooming.

The text is clear and easy to read.

The manuscript is well written and has a significant contribution to the field.

The design research is well described.

The literature consulted is varied and relatively recently.

The results are clearly presented.

Author Response

Thanks for your kind and positive comments on our manuscript.

Reviewer 2 Report

In this manuscript, the main parameters of oscillation extraction of flavonoids from bud of Daphne genkwa were explored by FFD, and the optimal extraction conditions were revealed by various analytical methods. The flavonoids in different flowering stages of Daphne genkwa were compared and analyzed. The changes of flavonoids in different flowering stages were revealed and three typical flavonoids were found. The work is interesting, which has important implications for the comprehensive study of flavonoid extraction and its changes in different flowering stages of Daphne Genkwa.

However, before it can be accepted for publication, the manuscript needs some clarifications and improvements.

  1. Generally speaking, Shaking time, Temperature and Shaking speed are important parameters in the actual extraction of chemical components. However, in this manuscript (Table 2), p-values of Time (Xa) and Shaking speed (Xc) reached 0.495 and 0.940 respectively, I think it is necessary for the author to analyze the reasons.
  2. It is necessary to design experiments to test the optimal extraction process conditions obtained by response surface methodology (RSM) analysis, however, in this manuscript, there is no information about the verification of optimal process conditions.

Generally, the language of paper should be improved throughout the text.

  1. Line 92-94: The expression of the sentence should be modified appropriately, in addition ‘for enrich’ should be ‘for enriching’, ‘best solvent’ should be ‘the best solvent’.
  2. Line 112: ‘the good’ should get rid of ‘the’.
  3. Line 134: The phrase ‘solvent mixture’ seems to be missing a determiner before it. There are similar problems elsewhere, for example, Line 141: ‘dependent variable’
  4. Line 149: ‘that may influence to the flavonoids extraction efficacy’ This sentence should be revised.
  5. Line 160: ‘the Xb and Xa Xc has striking positive influence to the flavonoids extraction efficacy’ should be ‘the Xb and Xa Xc had a striking positive influence on the flavonoids extraction efficacy’.
  6. Line 179: ‘mathematically’ should be ‘mathematical’.
  7. Line 203: ‘trace’ should be ‘traced’.
  8. Many single and plural forms in the manuscript also need to be checked and revised, for example, line 218: ‘stages’, ‘the VIP approach was used as variable selection method’, line 226: ‘exhibit’,
  9. Line 233-234: ‘The quality control of botanical sources is gradually progressing in the direction of identification and quantification all possible chemicals using fingerprint approach’ There's a missing 'of' before 'all'.

Author Response

In this manuscript, the main parameters of oscillation extraction of flavonoids from bud of Daphne genkwa were explored by FFD, and the optimal extraction conditions were revealed by various analytical methods. The flavonoids in different flowering stages of Daphne genkwa were compared and analyzed. The changes of flavonoids in different flowering stages were revealed and three typical flavonoids were found. The work is interesting, which has important implications for the comprehensive study of flavonoid extraction and its changes in different flowering stages of Daphne Genkwa.

However, before it can be accepted for publication, the manuscript needs some clarifications and improvements.

  1. Generally speaking, Shaking time, Temperature and Shaking speed are important parameters in the actual extraction of chemical components. However, in this manuscript (Table 2), p-values of Time (Xa) and Shaking speed (Xc) reached 0.495 and 0.940 respectively, I think it is necessary for the author to analyze the reasons.

: In general, shaking speed and extraction time have a significant effect on the extraction efficiency, but in the case of plant samples with fragile tissues such as the case of Genkwae Flos, these parameters do not seem to have a significant effect. This point was newly added to our revised manuscript.

  1. It is necessary to design experiments to test the optimal extraction process conditions obtained by response surface methodology (RSM) analysis, however, in this manuscript, there is no information about the verification of optimal process conditions.

: From the study of the optimization process, only one factor, temperature, was represented as an important factor, so optimization was conducted through a polynomial regression curve for that single factor.

 Generally, the language of paper should be improved throughout the text.

  1. Line 92-94: The expression of the sentence should be modified appropriately, in addition ‘for enrich’ should be ‘for enriching’, ‘best solvent’ should be ‘the best solvent’.

: It was corrected in Line 119 of the revised MS.

  1. Line 112: ‘the good’ should get rid of ‘the’.

: It was corrected in Line 128 of the revised MS.

  1. Line 134: The phrase ‘solvent mixture’ seems to be missing a determiner before it. There are similar problems elsewhere, for example, Line 141: ‘dependent variable’

: It was corrected in Line 159-172 of the revised MS.

  1. Line 149: ‘that may influence to the flavonoids extraction efficacy’ This sentence should be revised.

: It was corrected in Line 178 of the revised MS as follows: that have an important influence on the extraction efficiency of the shaking method.

  1. Line 160: ‘the Xb and Xa Xc has striking positive influence to the flavonoids extraction efficacy’ should be ‘the Xb and Xa Xc had a striking positive influence on the flavonoids extraction efficacy’.

: It was corrected in Line 193 of the revised MS.

  1. Line 179: ‘mathematically’ should be ‘mathematical’.

: It was corrected in Line 223 of the revised MS.

  1. Line 203: ‘trace’ should be ‘traced’.

: It was corrected in Line 237 of revised MS.

  1. Many single and plural forms in the manuscript also need to be checked and revised, for example, line 218: ‘stages’, ‘the VIP approach was used as variable selection method’, line 226: ‘exhibit’,

: The format of all singular and plural types in the manuscript has been reviewed and revised.

  1. Line 233-234: ‘The quality control of botanical sources is gradually progressing in the direction of identification and quantification all possible chemicals using fingerprint approach’ There's a missing 'of' before 'all'.

: It was corrected as suggested.

Reviewer 3 Report

Manuscript would benefit from a grammatical revision. There seem to be a number of instances where the wrong word is used i.e. quadratic when interaction is more likely correct. Some restructuring of information in the manuscript by moving it to a different location would improve quality of the paper.

Author Response

Manuscript would benefit from a grammatical revision. There seem to be a number of instances where the wrong word is used i.e. quadratic when interaction is more likely correct. Some restructuring of information in the manuscript by moving it to a different location would improve quality of the paper.

The manuscript requires significant grammar revision.

Daphne genkwa has three subspecies, genkwa, jinzhaiensis and leishanensis. Do you know which you were working with?

: As described in Material and Methods, the botanical origin of D. genkwa was confirmed through organoleptic examination.

On line 61 you indicate that you are working with 12 species of D. genkwa which does not seem possible. Please clarify as to subspecies or cultivar.

: ‘species’ was corrected to ‘trees’

Introduction

I would suggest moving Figure 4 into the introduction to demonstrate the flowering stages referenced in lines 55-58.

: This part was moved to the introduction section and Figure 4 was renumbered to Figure 1.

I would move the structure table from Figure 1 into the introduction to help the reader understand what flavonoids you were considering.

: This was moved to the introduction part and the original Figure 1 was renamed as Figure 2.

The first paragraph in the Results section could be moved into the introduction to help the reader understand what you are after.

: This part is a specific topic about the extraction, so we think it's better to leave it in the related result part.

The study examines solvent:shake time:heat interaction followed by life stage and this could be more clearly presented in the Introduction.

: It was revised in Introduction on lines 54-61 and line 85-98.

Results

For the equations don’t substitute X1, X2, X3 for Temperature, shaking time and solvent composition. See line 118.

: This equation is regarding the solvent mixture optimization and the meaning of each variable is correct.

Explain the importance of the interaction terms to the study. Lines 114-115. X1X2 is not a quadratic term but a interaction term. X1^2 would be a quadratic term. The interaction terms can be interpreted in the context of the study and a discussion of why an interaction term between temperature and shake time or solvent composition would matter.

: X1X2 was corrected as an interaction term and more discussion on the interaction between variables was added in the result and discussion section.

The identification of the compounds using TOF should be explained in the section associated with Figure 1. The details provide in Materials and Methods are not adequate to convince a reader that identification is correct.

: The UHPLC-TOF study on the identification of flavonoids in the extract of Genkwae flower buds was performed in our previous study and the description was added in the introduction part with citation.

Based on the design (Table 2) it seems reasonable to expect the evaluation of the models using Akaike Information Criteria corrected values to show that the most parsimonious model is being used.

: FFD factor selection study is a process of selecting important factors and it does not need to reveal a specific relationship with the responses, AICC seems not necessary.

Lines 124-125 Equation 1 is not quadratic. It is multivariate with interaction terms included.

: It was corrected as suggested.

The equation on line 177, the polynomial for temperature effects should be justified using AICC as it may be overfit. As a minimum show a first order and second order expressions with the corresponding AICC.

: Table 4 was newly added to provide the calculation results of the AICc and BIC values according to the first, second, and third regression equations, and the third regression equations showed the smallest AICc and BIC values, so they were selected and the corresponding description was also added on lines 208-220.

Figure 5 needs to be more fully explained. What percentage of variance is captured in the T1 and T2 terms in the PCA score(?) plot. Interpret this plot in the context of visually confirming the ability to classify the source of the extract based on flavonoid content. You might add a summary table for total peak areas with measures of variance for the three flower stages.

: Multivariate analysis did not represent percentages for T1 and T2 because OPLS-DA analysis was conducted rather than PCA analysis. From the Variable Importance in Projection (VIP) study, three flavonoids were selected as the most important metabolites responsible for this classification.   More discussion was added in the results section.

Figure 6 should include all 11 flavonoids assayed. The three could be then extracted out of the larger heat map to demonstrate the differences. Given the data available the use of a heat map instead of a 2-way ANOVA followed by a pairwise or other comparison should be justified. That the other 8 flavonoids may not be significantly different by flower stage should be discussed. Why the three were different across flower stage should be discussed in the context of other plants that show similar trends.

: Figure 6 was replaced with a new figure which showed the Heat-map for all 11 flavonoids and three significant flavonoids together. Some discussion regarding the comments was also added in the corresponding section.

Discussion

Line 242, explain how these three flavonoids are unique. Explain how they might be used as a quality control for freshness. This confounds the interpretation of your work. Are the predictive classifications of flower stage data impacted by quality and how so? How did you control for them in the study? What would constitute a measure of quality? These pints if included in the intro and then addressed throughout the manuscript would make this a very nice paper. As written the justification for this work is unclear.

: More discussion on this point was added in the Discussion section.

Reviewer 4 Report

The manuscript could be of interest for the journal readership. Data are quite interesting but presentation, English and discussion are very poor. I could not highlight all the phrases and the relative correction. The revision from a native English speaker is mandatory. I suggest resubmission after revision of the form and enrichment of the discussion.

Author Response

The English of the entire manuscript was revised by a native speaker, and more detailed discussion was added in Results and Discussion section.

Round 2

Reviewer 2 Report

The authors have addressed and revised most of the comments raised, and I thus recommend the acceptance of the work.

Reviewer 3 Report

Some grammar editing and consistent use of tense would make the manuscript easier to read. Much improved over the earlier version. 

Reviewer 4 Report

The quality of the presentation now reflects the scientific soundness.

The manuscript can be accepted